# Direct and Abscopal Antitumor Responses Elicited by AlPcNE-Mediated Photodynamic Therapy in a Murine Melanoma Model

**DOI:** 10.3390/pharmaceutics16091177

**Published:** 2024-09-06

**Authors:** José Athayde Vasconcelos Morais, Pedro H. A. Barros, Marcelo de Macedo Brigido, Clara Luna Marina, Anamelia Bocca, André de Lima e Silva Mariano, Paulo E. N. de Souza, Karen L. R. Paiva, Marina Mesquita Simões, Sonia Nair Bao, Luana C. Camargo, João P. Figueiró Longo, Amanda Alencar Cabral Morais, Ricardo B. de Azevedo, Marcio J. P. Fonseca, Luis A. Muehlmann

**Affiliations:** 1Laboratory of Nanoscience and Immunology, Faculty of Ceilandia, University of Brasilia Ceilandia Sul, Brasilia 72220-275, DF, Brazil; joseavmorais@gmail.com; 2Laboratory of Gene Regulation and Mutagenesis, Department of Genetics and Morphology, Institute of Biological Sciences, University of Brasilia, Brasilia 70910-900, DF, Brazil; 3Laboratory of Molecular Immunology, Department of Cell Biology, Institute of Biological Sciences, University of Brasilia, Brasilia 70910-900, DF, Brazil; pedro.henrique@aluno.com.br (P.H.A.B.); brigido@unb.br (M.d.M.B.); 4Laboratory of Applied Immunology, Institute of Biology Sciences, University of Brasilia, Brasilia 70910-900, DF, Brazil; clara.marina@unb.br (C.L.M.); albocca@unb.br (A.B.); 5Laboratory for Softwares and Physics Instrumentation Development, Institute of Physics, University of Brasilia, Brasilia 70910-900, DF, Brazil; demariano95@gmail.com (A.d.L.e.S.M.); psouza@unb.br (P.E.N.d.S.); 6Laboratory of Microscopy and Microanalysis, Department of Cell Biology, Institute of Biological Sciences, University of Brasilia, Brasilia 70910-900, DF, Brazil; karendepaiva@gmail.com (K.L.R.P.); marinamesquita3007@gmail.com (M.M.S.); snbao@unb.br (S.N.B.); 7Laboratory of Nanoscience and Nanobiotechnology, Department of Genetics and Morphology, Institute of Biological Sciences, University of Brasilia, Brasilia 70910-900, DF, Brazil; luanacrcamargo@gmail.com (L.C.C.); jplongo82@gmail.com (J.P.F.L.); amandaalencarcabral@gmail.com (A.A.C.M.); razevedo@unb.br (R.B.d.A.)

**Keywords:** immune system, immunity, immunotherapy, nanoemulsion, cancer, transcriptome

## Abstract

Melanoma, the most aggressive form of skin cancer, presents a major clinical challenge due to its tendency to metastasize and recalcitrance to traditional therapies. Despite advances in surgery, chemotherapy, and radiotherapy, the outlook for advanced melanoma remains bleak, reinforcing the urgent need for more effective treatments. Photodynamic therapy (PDT) has emerged as a promising alternative, leading to targeted tumor destruction with minimal harm to surrounding tissues. In this study, the direct and abscopal antitumor effects of PDT in a bilateral murine melanoma model were evaluated. Although only one of the two tumors was treated, effects were observed in both. Our findings revealed significant changes in systemic inflammation and alterations in CD4^+^ and CD8^+^ T cell populations in treated groups, as evidenced by blood analyses and flow cytometry. High-throughput RNA sequencing (RNA-Seq) further unveiled shifts in gene expression profiles in both treated and untreated tumors. This research sheds light on the novel antitumor and abscopal effects of nanoemulsion of aluminum chloride phthalocyanine (AlPcNE)-mediated PDT in melanoma, highlighting the potential of different PDT protocols to modulate immune responses and to achieve more effective and targeted cancer treatments.

## 1. Introduction

Melanoma, a highly aggressive form of skin cancer, represents a significant clinical challenge due to its high metastatic potential and recalcitrance to conventional treatments [1,2]. Despite advancements in surgical techniques, chemotherapy, and radiotherapy, the prognosis for advanced melanoma remains poor [1,2,3,4]. This fact spurs an intense search for novel therapeutic approaches that can more effectively combat this disease. Immunotherapy has emerged as a promising strategy, leveraging the immune system to target and eliminate tumor cells [3,4,5]. However, not all patients respond to it, and the efficacy of immunotherapy alone is often limited by the immunosuppressive tumor microenvironment and by the heterogeneous nature of melanoma, highlighting the need for complementary therapeutic strategies [3].

Photodynamic therapy (PDT) has drawn attention as an alternative, particularly for localized treatment with minimal damage to the surrounding healthy tissues [6]. PDT involves the generation of reactive oxygen species (ROS) following the activation of a photosensitizer by light at a wavelength corresponding to one of its absorbance bands, leading to tumor eradication through various mechanisms, including apoptosis, necrosis, vascular damage, and the induction of an antitumor immune response via immunomodulatory pathways. [7]. The ability of PDT to target primary tumors and to induce systemic antitumor immunity capable of addressing secondary tumors or metastases is a critical aspect of its therapeutic potential and has gained prominence in various studies [8].

Despite its potential, the antitumor efficacy and the ability of PDT to induce a robust immune response can be influenced by several factors, including the chemical nature and concentration of the photosensitizer, the light dose, the treatment regimen, and the wavelength of the applied light [9,10,11]. The wavelength is particularly crucial due to light scattering effects, which can limit the penetration of light and, consequently, the overall efficacy of the treatment. In this context, chloride aluminum phthalocyanine (AlPc) and its nanoemulsified form, AlPcNE, have emerged as potent photosensitizers for PDT. AlPcNE, in particular, exhibits significantly higher photodynamic activity in aqueous media compared to its non-nanoemulsified form, AlPc [12,13,14]. PDT mediated by AlPcNE has demonstrated efficacy against several types of tumors in preclinical studies [15,16] and also against bacteria [17] and fungi [18]. Simões and collaborators, 2024 [19] demonstrated that the lipid-based nanocarrier containing the photosensitizer aluminum phthalocyanine chloride (SLNs-AlPc), when light-activated, increased ROS production in B16F10 melanoma cells, modulated the dendritic cell profile, and induced cell death accompanied by DAMP exposure and autophagosome formation. Furthermore, Mkhobongo et al. (2022) [20] demonstrated that PDT with a gold nanoparticle–aluminum phthalocyanine conjugate significantly increased cytotoxicity and apoptosis while drastically reducing the proliferation and viability of melanoma cancer stem cells, compared to the non-conjugated treatment and controls. However, the potential for abscopal effects and the impact on immune system gene expression in melanoma treatment still require further investigation.

This study assessed the antitumor efficacy and the immune-modulating effects of AlPcNE-mediated PDT in a murine model of melanoma (B16F10). The direct effects on treated tumors and the systemic, abscopal effects on distant, untreated tumors, immune cell modulation, and gene expression profiles were investigated.

We aimed to gain insights into the interplay between PDT and the immune system, which could contribute to more effective and targeted melanoma treatments and improved outcomes for patients battling this kind of cancer.

## 2. Materials and Methods

### 2.1. Cell Culture

The murine melanoma cell line B16F10 was purchased from the Rio de Janeiro Cell Bank (BCRJ, Rio de Janeiro, Brazil). Cells were cultured in Dulbecco’s Modified Eagle Medium containing 10% fetal bovine serum and 1% penicillin/streptomycin at 37 °C in a 5% CO_2_ atmosphere.

### 2.2. Animals

All experiments involving mice were approved by the Animal Ethics Committee of the University of Brasilia (UnBDOC n◦ 46/2019). Female, 18 ± 2 g, 6- to 8-week-old C57Bl/6 mice (Faculty of Medicine—Federal University of Goiás, Goiania, GO, Brazil) were maintained in a temperature-controlled environment with 12 h light–dark cycles and received food and water ad libitum.

### 2.3. Tumor Model

To assess the direct and abscopal PDT effects, a bilateral tumor model was used. Mice were injected s.c. with 5 × 10^5^ B16F10 cells in the right flank (primary tumor). Two days later, 2 × 10^5^ tumor cells were implanted in the left flank (secondary tumor). Tumor size was monitored by caliper measurement every two days. Tumor volume was calculated as follows: tumor volume (mm^3^) = width (mm) × width (mm) × length (mm)/2 (Figure 1).

### 2.4. Photodynamic Therapy Protocols

The nanoemulsion of aluminum phthalocyanine chloride (AlPcNE) was prepared using the spontaneous emulsification method as described by Muehlmann et al. (2015) [12]. The resulting AlPcNE formulation was thoroughly characterized, exhibiting a mean hydrodynamic diameter (DH) of 27.15 ± 1.070 nm, a polydispersity index (PdI) of 0.178 ± 0.026, and a zeta potential (ZP) of −0.340 ± 0.188 mV. The final concentration of aluminum phthalocyanine chloride in the nanoemulsion was determined to be 40 µM. For the PDT treatment, mice were intratumorally injected with 200 µL of different AlPcNE concentrations into the right tumors on day 12. Thirty minutes after injection, the animals were anesthetized and covered with aluminum foil, leaving only the right tumor area exposed for LED irradiation. A custom-designed light-emitting diode (LED) device, developed by Prof. Paulo Eduardo Narcizo de Souza and configured to deliver light at a wavelength of 660 nm, was used to irradiate the animals. The LED array was positioned at a fixed distance of 10 cm from the tumor surface to ensure consistent fluence. Mice were randomly assigned to be treated only on the primary tumor with either: PBS, (1) low energy density [25 J/cm^2^] with low AlPcNE concentration [10 nM] (LL_LPS); (2) low energy density [25 J/cm^2^] with high concentration of AlPcNE [40 nM] (LL_HPS); (3) high energy density [112 J/cm^2^] with low AlPcNE concentration [10 nM] (HL_LPS); or (4) high energy density [112 J/cm^2^] with high concentration of AlPcNE [40 nM] (HL_HPS). As control, mice were intratumor injected with PBS on day 12. Mice were euthanized on day 18.

### 2.5. Hematological Parameters Analysis

On the day of euthanasia, the animals were anesthetized by 120 mg/kg ketamine and 16 mg/kg xylazine. Blood samples were then collected by cardiac puncture in EDTA Vacutte^®^ microtubes. The hematological parameters of WBC (white blood cells), RBC (red blood cells), HGB (hemoglobin), HCT (hematocrit), PLT (platelets), MCV (mean corpuscular volume), MCH (mean corpuscular hemoglobin), and MCHC (mean corpuscular hemoglobin concentration) were evaluated by the automated hematology counter for veterinary use, Horiba ABX Micros ESV 60 (São Paulo, Brazil).

### 2.6. Histology Analysis

On day 18, mice were euthanized and dissected to collect the irradiated and non-irradiated tumors. The tissue samples were carefully collected and immersed in 10% neutral-buffered formalin for 24 h to prevent autolysis. After fixation, the tissue samples were dehydrated in a series of graded ethanol solutions (70%, 80%, 90%, and 100%). Following dehydration, the samples were cleared in ethanol–xylene and xylene for 1 h each, then infiltrated with molten paraffin wax (56–58 °C) for 2–4 h. The tissues underwent three paraffin baths in an oven at 58 °C and were embedded in paraffin blocks. Histological sections, 3 to 4 μm thick, were fixed on glass slides for microscopy and stained with hematoxylin and eosin (HE).

### 2.7. Flow Cytometry Analysis

The single-cell suspension of mice splenocytes was prepared by mechanical digestion followed by filtering through a 40 mm cell strainer (SPL Life Sciences, Pocheon, Republic of Korea). Cells samples were resuspended in erythrolysis buffer for 10 min, washed with PBS, centrifuged, and counted. Cells were transferred to polypropylene tubes in 100 μL PBS + 10% FBS for labeling for 30 min on ice, protected from light, with the Invitrogen APC anti-mouse CD3ε, PE/Cyanine7 anti-mouse CD4, and PE anti-mouse CD8b antibodies. The FITC anti-mouse CD45 and PerCP Cy5.5 anti-mouse CD45RA antibodies were used for activated T cells Panel 1. The FITC anti-mouse CD25 and PerCP Cy5.5 anti-mouse CD62L antibodies were used for regulatory T cells Panel 2. After additional washes with PBS containing 2% FBS, the cells were resuspended, acquired by FACSVerse (BD Biosciences, San Jose, CA, USA), and analyzed by FlowJo software X (version 10.10, BD Biosciences). Cells labeled with individual antibodies were used for compensation.

### 2.8. Gene Expression Evaluation

The total RNA was isolated from the frozen homogenized independent samples of melanoma primary and secondary allografts (n = 3 per group) using the RNeasy Mini kit (Qiagen, Hilden, Germany), according to the manufacturer instructions. Samples were quantified by the RNA-specific fluorimetric method using the Qubit™ RNA High Sensitivity kit (Invitrogen™, Waltham, MA, USA), following the manufacturer’s recommendations. Sequencing was carried out by the company GenOne Biotech using the Illumina platform (Illumina, Inc., San Diego, CA, USA). According to the company’s requirements, only samples with pure (OD 260/280 > 2.0) and intact (with RIN > 6.3) total RNA were sent for sequencing. Samples should consist of at least 2 µg of lyophilized RNA.

### 2.9. Statistical Analysis

All the statistical differences were recorded using GraphPad Prism version 6.01 for Windows (San Diego, CA, USA). Statistical analyses were assessed by one-way or two-way ANOVA tests, with post hoc Tukey’s multiple-comparison tests. Values are presented as mean ± standard error of the mean. Significance was set at *p* < 0.05.

## 3. Results

### 3.1. The PDT Protocols, except LL_LPS, Reduced the Growth of the Irradiated Tumor

To verify the in situ effect of AlPcNE-PDT on melanoma grafts, four different PDT protocols were applied. As shown in Figure 2A,B, in a bilateral B16F10 model, AlPcNE-PDT was administered locally only to the primary tumor, which was monitored for 18 days.

The LL_HPS, HL_LPS, and HL_HPS protocols significantly reduced the primary-site tumors’ growth. In contrast, no direct effect was observed with the LL_LPS protocol, as the tumors continued to grow similarly to the control group. This could be due to the lower oxidation of cell components in the tumors treated with the LL_LPS protocol, as it involved lower concentrations of both AlPcNE and light energy.

### 3.2. Histological Analyses of the Primary-Site Tumors

Typical features of melanoma tissue were verified in all the samples (Figure 3). Notably, atypical cells were present, characterized by pleomorphic nuclei. Areas of multifocal necrosis were evident (eosin stained—arrow), displaying regions of cellular death, with absence of cell nucleus and no intercellular limits, indicative of an aggressive and rapidly proliferating tumor microenvironment. Additionally, the analysis confirmed the presence of pigment characteristic of melanoma cells (*), reasserting the melanocytic origin of the tumor. Importantly, tumors in the PBS group exhibited a higher frequency of cells in mitosis compared to the treated groups, suggesting increased cellular proliferation.

### 3.3. The Growth of the Non-Treated Tumors of the HL_HPS Group Is Significantly Reduced after PDT

The abscopal effect of the four different AlPcNE-PDT protocols on melanoma grafts was also verified. The secondary-site tumor was left untreated. Only the HL_HPS PDT protocol group presented significant growth reduction of the secondary-site tumor when compared to the other groups (Figure 4A,B), suggesting the induction of systemic antitumor responses against B16F10 cells.

Even though the group LL_HPS and HL_LPS PDT protocols directly reduced the primary-site tumors’ volume, no induction of a systemic antitumor activity was detected. Abscopal antitumor activity was not promoted by the application of the LL_LPS protocol either.

### 3.4. Histological Analyses of the Secondary-Site Tumors

The histological analysis of the secondary-site tumors revealed typical features of melanoma, like the primary tumor (Figure 5). Atypical cells, characterized by pleomorphic nuclei typical of melanoma, were observed. Unlike the primary tumor, the secondary-site tumors displayed multiple small areas of necrosis (stained with eosin), rather than large, focal areas. The presence of the pigment characteristic of melanoma cells was also noted.

### 3.5. Hematological Analysis

To investigate how AlPcNE-PDT protocols influenced hematological parameters in mice with B16F10 grafts, blood samples were collected on the day of euthanasia and evaluated. All the AlPcNE-PDT protocols led to a decrease in the number of circulating leukocytes and lymphocytes compared to the untreated group (Table 1). The HL_HPS groups were associated with lower platelet levels. The LL_LPS and HL_HPS protocols showed a reduction in the number of circulating HCT. The levels of neutrophils, monocytes, eosinophils, and basophils remained unaffected by AlPcNE-PDT.

### 3.6. Flow Cytometry Analyses

To evaluate the T cell population in mice spleens, the frequency of CD4+ T and CD8+ T cells was analyzed by flow cytometry. In mice from groups LL_LPS and HL_HPS, the population of CD4+ T cells decreased compared to the control groups (Figure 6A). Additionally, the splenic CD8+ T cell population in HL_HPS-treated mice increased compared to the controls (Figure 6B). These findings suggest that treatment with HL_HPS may promote antitumor immunity through both CD4+ and CD8+ T cells. There were no statistically significant differences in the CD4+ and CD8+ T cell populations between groups LL_HPS and HL_LPS compared to the control.

To explore the subpopulation of splenic CD4+ and CD8+ T cells, markers for CD45+ and CD45RA+ (Panel 1) and CD25+ and CD62L+ cells (Panel 2) were used.

Significant differences were revealed in the mean fluorescence intensity (MFI) of CD4+CD45+ cells among the experimental groups. Specifically, the HL_HPS protocols significantly increased the percentage of CD4+CD45+ cells compared to control. No statistically significant differences of CD4+CD45+ cells were observed between LL_LPS, LL_HPS, and HL_LPS groups and PBS (Figure 7A). These results suggest that the HL_HPS treatment may induce higher differentiation of CD4+CD45+ T cells compared to the control group, while the LL_LPS, LL_HPS, and HL_LPS protocols do not.

The LL_HPS group exhibited fewer CD8+CD45+ cells compared to the HL_HPS group. No statistically significant differences of CD8+CD45+ cell frequencies were observed for the LL_LPS and HL_LPS protocols compared to the control group (Figure 7C).

Collectively, these findings indicate that the HL_HPS protocol can promote a T-cell-specific immune response.

### 3.7. The HL_HPS Protocol Induced Distinct Expression Profiles for Immune Response Genes in Primary- and Secondary-Site Tumors

The reduction in the growth of the non-treated tumor, together with the distinct immunological responses elicited by the HL_HPS_PDT protocol, was the most significant finding in this study. Consequently, we concentrated on the transcriptomic analysis via RNA-Seq of primary- and secondary-site tumors in irradiated animals from the HL_HPS condition, in comparison with tumors from the PBS group. This approach aimed to elucidate and correlate the underlying antitumor mechanisms involved.

For RNA sequencing, three independent samples from both primary- and secondary-site tumors of the PBS and HL_HPS groups were collected. These samples met the required criteria for total RNA concentration, quality, and purity stipulated by the service provider. Gene expression levels were estimated by transcript abundance, with values normalized to FPKM (fragments per kilobase of transcript per million mapped reads), accounting for sequencing depth and gene length effects on the fragment counts.

The number of unique and co-expressed genes within each PDT and PBS protocol was illustrated using a Venn diagram. In the right tumor samples, 11,658 genes were co-expressed, with 284 and 415 unique genes in the PBS and HL_HPS group, respectively. In contrast, the left tumors presented 11,755 genes co-expressed across the two groups, with 738 and 412 unique genes in the PBS and HL_HPS group, respectively (Figure 8).

Profound alterations in gene expression profiles were observed between irradiated and control animals (Figure 9). Additionally, there was a noticeable similarity in gene expression profiles between the right and left tumors in the PBS group, whereas the right and left tumors in the HL_HPS group exhibited predominantly different profiles.

Genes with significantly different expression levels under various conditions (|log2(FoldChange)| ≥ 1 and padj ≤ 0.05) were identified. A log2FC > 0 indicates higher gene expression in the experimental condition compared to the control condition, whereas a log2FC < 0 indicates lower expression in the experimental condition compared to the control.

The global distribution of differentially expressed genes in the HL_HPS-PDT protocol for the primary- and secondary-site tumors compared to the PBS control group, as well as between primary and secondary tumors within the same group, was inferred by volcano plots (Figure 10).

Comparing the PBS_R and HL_HPS_R samples, a total of 693 differentially expressed genes were identified, with 273 upregulated and 420 downregulated transcripts in HL_HPS_R. The PBS_L vs. HL_HPS_L sample comparison revealed 1208 differentially expressed genes, including 835 upregulated and 373 downregulated in HL_HPS_L. The comparison of PBS_R and PBS_L samples identified 1052 differentially expressed genes, with 100 upregulated and 952 downregulated in PBS_L. The HL_HPS_R vs. HL_HPS_L sample comparison revealed 121 differentially expressed genes, including 60 upregulated and 61 downregulated sequences in HL_HPS_L.

The evaluation of the gene ontology of DEG in the PBS_R vs. HL_HPS_R sample comparison identified several genes related to the immune response, most of them being downregulated (Figure 11A). Conversely, in the PBS_L vs. HL_HPS_L sample analysis, differentially expressed immune-response-related genes were predominantly upregulated (Figure 11B).

Table 2 highlights the DEGs for the comparisons PBS_R vs. HL_HPS_R and PBS_L vs. HL_HPS_L based on the GO category for Immune System Processes (GO:0002376). This analysis revealed that the HL_HPS_R tumor exhibited 28 upregulated genes and 60 downregulated genes associated with immune system processes, while in the HL_HPS_L tumor, 122 genes were upregulated and 49 were downregulated.

It was possible to identify genes that are upregulated both in the HL_HPS_R and HL_HPS_L tumors compared to the same tumors in the PBS group, such as Herc6, Kat7, Mef2c, Rbm15, and Shld3 (Table 2). On the other hand, the genes that were downregulated both in HL_HPS_R and HL_HPS_L tumors compared to the same tumors in the control group were Cdc42ep2, Cebpg, Fzd9, Gprc5b, Hexim1, Kcnj8, Mitf, Nfkb2, Nr4a3, Prkd2, Rarg, Tbkbp1, Vegfa, Xkr8, and Zbtb7b (Table 2).

The comparison of DEGs between the primary- and secondary-site tumors in each treatment indicated that, in the PBS group, most immune-response-related genes were downregulated in the secondary-site tumor compared to the primary one (Figure 12A). In contrast, the primary- and secondary-site tumors in the HL_HPS group did not exhibit significant differential expression of immune-related genes (Figure 12B).

Table 3 presents the DEGs identified in the comparison between tumors within the same group, PBS_R vs. PBS_L, and HL_HPS_R vs. HL_HPS_L. This analysis revealed 12 upregulated and 206 downregulated genes associated with immunological processes in PBS_L. Conversely, in the HL_HPS_L group, only seven genes related to immunological processes were upregulated and three were downregulated.

Overall, The HL_HPS-PDT protocol elicited substantial changes in the gene expression profiles of both primary- and secondary-site tumors, in comparison to the corresponding tumors of the PBS group. These alterations underscore the impact of HL_HPS-PDT on the molecular landscape of the tumors, highlighting its potential as an effective modulator of tumor gene expression.

## 4. Discussion

In recent years, photodynamic therapy (PDT) has garnered increasing attention due to its broad range of anticancer mechanisms [9,13,14,21]. The impact of PDT on the immune system is becoming an area of intense study [22,23]. The results of this study highlight the efficacy of PDT in treating grafted melanoma in mice, emphasizing significant effects on both the primary-site and distant tumors (abscopal effect), providing insights into tumor and immune-response mechanisms. To test the direct and abscopal effects mediated by AlPcNE-PDT, a bilateral melanoma graft model (B16F10) in mice was established, with PDT applied to only one of the tumors.

The LL_HPS, HL_LPS, and HL_HPS PDT protocols showed a significant reduction in the volume of the primary-site tumor. This effect can be attributed to the antitumor capabilities of PDT through the production of reactive oxygen species (ROS), leading to necrosis and apoptosis of tumor cells. Kolarova et al. (2007) [24] demonstrated that PDT can inhibit the proliferation of malignant melanoma cells by increasing intracellular ROS levels.

Notably, the HL_HPS condition not only reduced the volume of the primary-site tumor but also induced an abscopal effect, evidenced by the reduction in the volume of the secondary-site, non-irradiated tumor grafted on the opposite side. This phenomenon suggests the activation of a systemic immune response capable of attacking tumors distant from the primary treatment site. This abscopal effect is particularly significant as it indicates a therapeutic potential for treating metastases in locations not directly exposed to PDT. A similar effect was found in other studies, such as Gurung et al. (2023) [25], which showed that PDT with Ce6 as a photosensitizer was able to induce potent local and systemic antitumor immune responses in a murine model of malignant melanoma, further enhanced by the combination with PD-1/PDL-1 inhibitors.

The variation in efficacy among treatments can be attributed to differences in the concentration of the photosensitizer, as well as to the different light dosages. According to Morais et al. (2021) [9], different PS concentrations in PDT can induce different biological responses, including various types of cell death and the release of DAMPs, thereby influencing the induction of immunogenic cell death. Udartseva and colleagues (2019) [26] found that low-dose PDT significantly increased the secretion of proangiogenic factors such as VEGF-A, IL-8, PAI-1, and MMP-9 by in vitro mesenchymal stem cells (MSCs) and enhanced their angiogenic potential, suggesting a heightened pro-tumorigenic capability of MSCs. In contrast, Doix et al. (2019) [27] reported that low doses of the non-porphyrin photosensitizer OR141 were found to more effectively induce damage-associated molecular patterns (DAMPs) in vitro and to suppress the growth of squamous cell carcinoma in mice compared to higher doses. This supports our study finding that mice treated with different PS concentrations and light dosages exhibited varying direct and abscopal antitumor outcomes.

The observed hematological changes highlight the impact of different photodynamic therapy (PDT) protocols on systemic physiology. The PBS control group shows a WBC count of 10.8 ± 2.7 × 10^3^/μL, which is within the normal range for C57BL/6 mice (typically around 6.0−10.0 × 10^3^/μL) as reported in studies like White et al. (2016) [28]. The elevated WBC levels in the PBS group could indicate a systemic inflammatory response to the grafted tumor, corroborating the chronic inflammatory responses seen in tumor-bearing hosts [29]. Conversely, the LL_LPS, LL_HPS, and HL_LPS, groups showed significantly lower WBC counts, particularly the HL_HPS group with 4.4 ± 1.3 × 10^3^/μL, suggesting a reduction in systemic inflammation in response to the treatment.

Lymphocyte (LYM) counts also varied, with the PBS group showing elevated levels (6.2 ± 1.3 × 10^3^/μL), which aligns with chronic immune responses described for tumors [30]. Treated groups showed reduced LYM counts, with HL_HPS exhibiting 2.7 ± 1.4 × 10^3^/μL, suggesting a dampening of the immune response or a shift towards immune regulation. According to Was et al. (2020) [31], this reduction may indicate an active mobilization of leukocytes from the blood to the tumor site.

Additionally, the reduction of WBC and LYM counts in treated groups suggests that PDT may mitigate systemic inflammation, which could contribute to the antitumor efficacy by reducing pro-tumorigenic inflammatory environments. This can be explained by the modulation of the immune response, promoting the activation of dendritic cells and the presentation of tumor antigens, which may lead to a more effective and targeted immune response, resulting in the reduction of WBC and LYM levels [32]. Other studies also report that PDT can decrease inflammatory markers in animal models and patients [33].

Studies have demonstrated that the presence of tumors can trigger a chronic immune response, characterized by the increased presence of leukocytes and lymphocytes in peripheral blood. This response reflects the immune system attempt to combat the tumor presence, although it is often insufficient to contain tumor growth [34,35]. On the other hand, this inflammation can create a pro-tumor environment that favors cancer cell survival and invasion, increased angiogenesis, and recruitment of immunosuppressive cells, allowing tumor cells to escape immune destruction [34]. Studies have shown that inflammatory cytokines such as TNF-α, IL-6, and IL-1β can activate cellular signaling pathways that promote tumor survival and growth [36]. Understanding these hematological impacts is crucial for optimizing PDT regimens’ therapeutic benefits while minimizing adverse effects.

The present study found that AlPcNE-PDT induced a systemic increase in the population of cytotoxic CD8^+^ T cells and a reduction in the population of CD4^+^ T cells in the HL_HPS group. The reduction in CD4^+^ T cells may reflect an impact on immune regulation, while the increase in CD8^+^ T cells suggest cytotoxic activation against tumor cells. It is well known that subpopulations of CD4^+^ T cells, such as regulatory T cells (Tregs), promote immunosuppression, while helper T cells (Th) support CD8^+^ T cells by providing activating cytokines [37]. Our study suggests that, despite the reduction in the systemic population of CD4^+^ T cells, most of these cells in the HL_HPS group correspond to the differentiated CD4^+^CD45^+^ form, an important marker of antigen receptor signal transduction and lymphocyte development, indicating a potential long-term immune response [38]. CD8^+^ T cells, which have effector cytotoxic functions, are known to slow down distal tumor growth [39,40].

Several studies have demonstrated that PDT stimulates the immune system in various ways, including the release of tumor-associated antigens (TAAs) and immunostimulatory molecules from tumors, which can activate and trigger an anticancer immune response [41,42]. While PDT has been shown to activate both humoral immunity and cell-mediated adaptive immunity, CD8^+^ T cells are primarily responsible for the immunological effects of PDT [41,42]. Our findings demonstrated that the systemic population of CD8^+^ T cells was enhanced by AlPcNE-PDT in the HL_HPS condition. However, systemic T cells were depleted in the other irradiated groups and were unable to evoke an anticancer immune response. This could explain the non-occurrence of an abscopal effect with PDT, in groups other than HL_HPS.

Despite the direct antitumor effect observed in the LL_HPS and HL_LPS groups, antitumor immunity appears to be insufficient to eradicate the melanoma. This is possibly due to the low proliferation and activation of CD8^+^ T cells in the face of the tumor’s aggressiveness. According to previous studies, in the tumoral microenvironment, barriers can develop to prevent immune cells from migrating and penetrating the non-irradiated tumor [40,43].

Encouraged by the immunogenic capabilities of HL_HPS-PDT, we hypothesized that the antitumor immunity generated resulted from the modulation of the expression of genes involved in immune system processes. RNA-Seq data revealed distinct patterns of gene expression in response to the treatment. In the HL_HPS condition, the primary-site tumor exhibited more downregulated genes related to immunological processes, while the secondary-site tumor presented prevalence of upregulated genes in the same category. This suggests that the mechanisms of tumor growth inhibition were different in the two tumors.

The genes Herc6, Kat7, Mef2c, Rbm15, and Shld3 were found to be upregulated in both the right and left tumors of the treated groups, suggesting a coordinated response in tumor inhibition. Mao and colleagues (2018) [44] and Sala-Gaston et al. (2020) [45] proposed that HERC proteins may have dual roles in cancer, acting as oncogenes or tumor suppressors depending on the tumor type, although there is a lack of studies specifically on HERC6. Swanson and collaborators (1998) [46] demonstrated that all members of the MEF2 family (A–D) are expressed in B cell lines and all, except MEF2C, are expressed in T cell lines. This suggests that MEF2C expression is related to B cell development and function. Newman and colleagues (2017) [47] highlighted the essential role of Kat7 in the development, fitness, and survival of T cells, particularly in maintaining the acetylation of the histone 3 lysine 14 (H3K14ac), an essential epigenetic mark for the development of a normal immune system. Gao and collaborators (2021) [48] identified KAT7 as a tumor suppressor protein in colorectal cancer and non-small-cell lung cancer, while Dong et al. (2023) [49] showed that Rbm15 is associated with pancreatic cancer progression by promoting tumor proliferation, migration, and metastasis. Nonetheless, the role of Rbm15 on melanoma cells remains understudied. Lastly, Shld3 is involved in DNA regulation and repair [50].

The high expression of CD3D correlates with immune cell infiltration and with the response to immunotherapy in patients with head and neck squamous cell carcinoma [51] and colon adenocarcinoma [52]. Upregulation of CXCL10 in the HL_HPS_R tumor and upregulation of CXCL11 in the HL_HPS_L tumor seem relevant since they act as key immunological chemoattractant during inflammatory responses [53]. The positive modulation of these genes can potentially enhance the efficacy of the immune response against melanoma tumor cells.

Calabrese and co-workers (2009) [54] showed that the inactivation of SOCS1 disabled the p53-dependent senescence in response to oncogenic STAT5A and radiation-induced apoptosis in T cells, corroborating the results of inhibition of tumor growth and the upregulation of this gene in HL_HPS_R and indicating that SOCS1 could be a key gene in inhibiting tumor growth.

Another significant gene found as a DEG of the primary-site tumor in the HL_HPS group is CRTAM. CRTAM is a protein-coding gene expressed on the surface of activated NK T cells and CD8^+^ T lymphocytes. It enhances the infiltration into the tumor of immune cells, particularly CD8^+^ T cells. Moreover, CRTAM may promote the proliferation of activated T cells and the secretion of interferon (IFN)-γ, thereby enhancing the antitumor effectiveness of T cells [55].

Huang and colleagues (2020) [56] found a strong positive correlation between the gene expression of CXCL10 and the infiltration into the tumor of immune cells (B cells, CD8^+^ T cells, CD4^+^ T cells, macrophages, neutrophils, dendritic cells).

Several genes associated with memory, activation, and survival of CD8 T cells, such as REL and FOXP1, were found to be upregulated in the secondary-site HL_HPS tumor, in accordance with Feldman et al. (2018) [57], in the melanoma mice model.

These results suggest a differential modulation of immune responses, potentially reflecting a systemic immune activation.

Among the downregulated genes known to play an important role in melanoma progression are CXCL1, which exerts melanoma growth-stimulating activity, and MITF, a crucial oncogenic transcription factor to maintain tumor survival, increase proliferation, and promote differentiation [58,59,60]. These downregulated genes are involved in the signaling pathway and in the activation of the immune system. They also reduce immunosuppression and decrease tumor cell survival and proliferation. The genes CXCL1 and MITF were downregulated in both tumors of the HL_HPS group. This downregulation could create an environment less favorable to tumor growth and more permissive to the action of immune cells, resulting in a more effective AlPcNE-PDT response and the potential increase in abscopal effects.

Comparison between primary- and secondary-site tumors in the control group revealed a higher number of downregulated genes in the secondary one, indicating an immunosuppressive or adaptive response. In contrast, the absence of significant DEGs between primary- and secondary-site tumors in the AlPcNE-PDT HL_HPS-treated group suggests a homogenization of the immune response, likely due to the dissemination of systemic immune signals induced by AlPcNE-PDT.

The noteworthy results in the HL_HPS group can be attributed to the ability of AlPcNE-PDT to modulate the expression profile of genes involved in the immune system process. This modulation led to an increase in the systemic population of CD8^+^ T cells and the identification of an abscopal effect in the non-irradiated tumor.

## 5. Conclusions

Our findings suggest for the first time that AlPcNE-PDT can induce potent local and systemic antitumor immune responses. The antitumor effects of AlPcNE-PDT were achieved through direct action and by modulation of the gene expression profile of both primary- and secondary-site tumors, leading to the observed abscopal effects. To our knowledge, this is also the first evaluation of AlPcNE-PDT impact on both direct and abscopal effects.

This study confirmed the efficacy of PDT in reducing tumor volume and inducing systemic immune responses, particularly highlighted in the HL_HPS-PDT group, which presented an abscopal effect. Hematological changes and alterations in the T cell population indicated a complex interaction between PDT and the immune system, suggesting new avenues for optimizing combination therapies. RNA-Seq analysis reinforces the idea that PDT can modulate gene expression related to the immune system in a localized and systemic manner. To strengthen our conclusions, it would be beneficial to conduct additional studies to validate the RNA-Seq results, using techniques such as RT-PCR, and to investigate the molecular mechanisms underlying the abscopal effect.

In summary, this study demonstrated that PDT with AlPcNE can induce local and systemic antitumor immune responses, representing a potentially robust approach to enhancing melanoma treatment outcomes. The identification of modulated genes by the therapy, in both primary- and secondary-site tumors, suggests new therapeutic targets and strategies to enhance the abscopal effects and reduce the adverse effects. These discoveries represent a significant advancement in the field of photodynamic therapy and may positively impact melanoma and other cancer treatments.

## Figures and Tables

**Figure 1 pharmaceutics-16-01177-f001:**
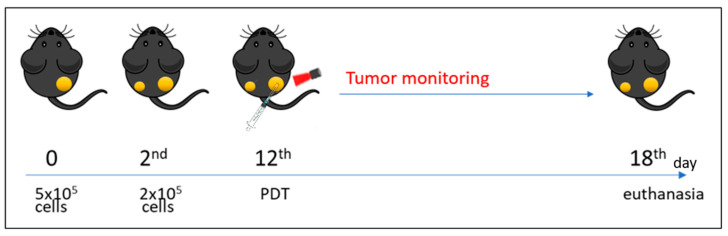
Treatment scheme for AlPcNE-mediated PDT in the mice animal model.

**Figure 2 pharmaceutics-16-01177-f002:**
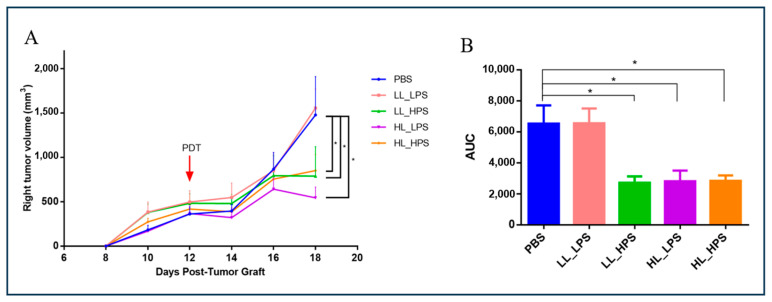
Direct effect of PDT in primary-site tumors. Primary subcutaneous B16F10 tumor-bearing mice were treated with four different PDT protocols: low LED with low PS (LL_LPS), laser irradiation at 25 J/cm^2^, and AlPcNE (10 nM); low LED with high PS (LL_HPS), laser irradiation at 25 J/cm^2^, and AlPcNE (40 nM); high LED with low PS (HL_LPS), laser irradiation at 112 J/cm^2^, and AlPcNE (10 nM); and high LED with high PS (HL_HPS), laser irradiation at 112 J/cm^2^, and AlPcNE (40 nM). (**A**) Tumor volumes (mm^3^) for irradiated and PBS right tumors are shown. (**B**) The area under the curve (AUC, expressed in mm^3^·day) was analyzed on day 18 post-tumor-engraftment to assess the cumulative tumor growth over time. The AUC provides an integrated measure of tumor volume, taking into account both the magnitude and duration of the tumor growth response. Data are mean ± standard deviation. Statistical significance of the tumor volume was determined using a two-way ANOVA, followed by a Tukey’s multiple-comparison test (n = 5, * *p* < 0.01).

**Figure 3 pharmaceutics-16-01177-f003:**
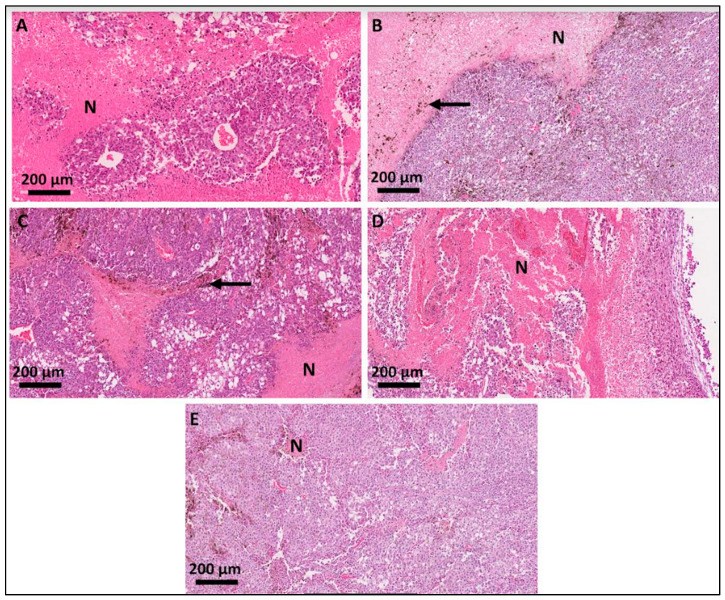
Photomicrographies of histopathological sections of the mice primary-site tumors. (**A**) PBS; (**B**) LL_LPS; (**C**) LL_HPS; (**D**) HL_LPS; (**E**) HL_HPS groups. N represents necrosis areas.

**Figure 4 pharmaceutics-16-01177-f004:**
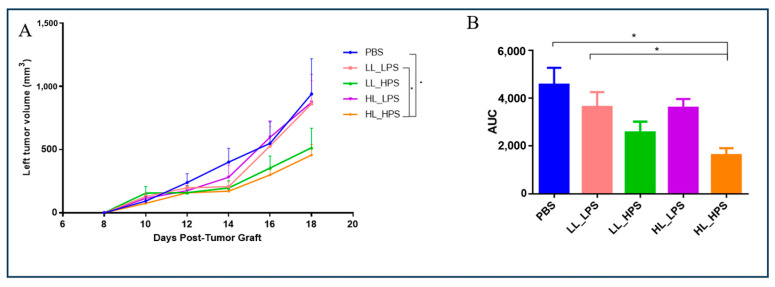
Abscopal effect of non-irradiated left tumors. (**A**) Volume growth (mm^3^) of the secondary-site tumors is shown. (**B**) The area under the curve (AUC, expressed in mm^3^·day) was analyzed on day 18 post-tumor-engraftment to assess the cumulative tumor growth over time. The AUC provides an integrated measure of tumor volume, taking into account both the magnitude and duration of the tumor growth response. Data are mean ± standard deviation. Statistical significance of the tumor volume was determined using a two-way ANOVA, followed by a Tukey’s multiple-comparison test (n = 5, * *p* < 0.01).

**Figure 5 pharmaceutics-16-01177-f005:**
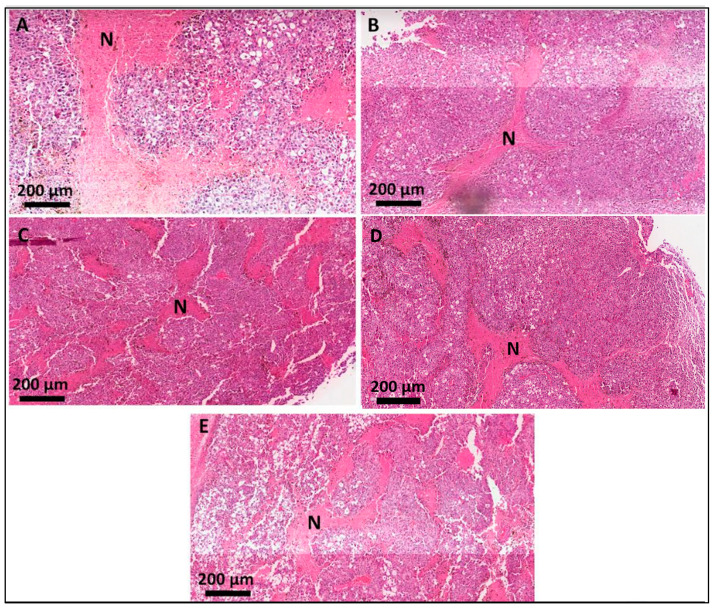
Photomicrographies of histopathological sections of the mice secondary tumor sites. (**A**) PBS; (**B**) LL_LPS; (**C**) LL_HPS; (**D**) HL_LPS; (**E**) HL_HPS groups (Scale bar—200 µm). Stained using H&E (hematoxylin and eosin).

**Figure 6 pharmaceutics-16-01177-f006:**
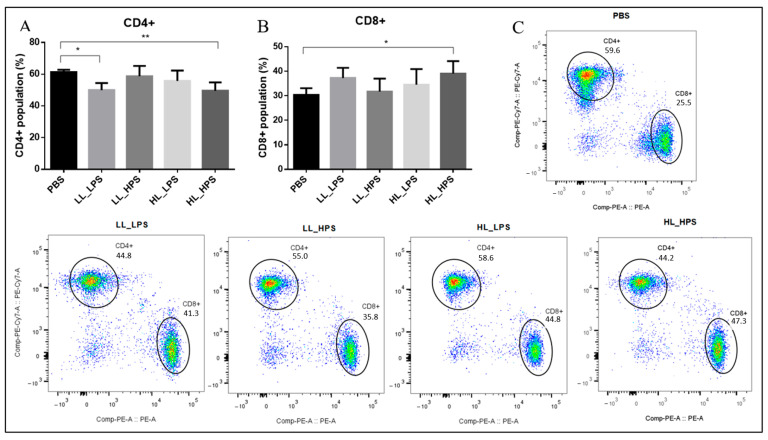
Flow cytometry analyses of the mice spleens’ T cell populations. (**A**) Percentage of CD4+ T cells; (**B**) percentage of CD8 + T cells. (**C**) CD4+ and CD8+ T cells. (n = 6). One-way ANOVA followed by Tukey’s post hoc test was performed for statistical analysis; * *p* < 0.05, ** *p* < 0.01. Data are presented as the mean ± SD.

**Figure 7 pharmaceutics-16-01177-f007:**
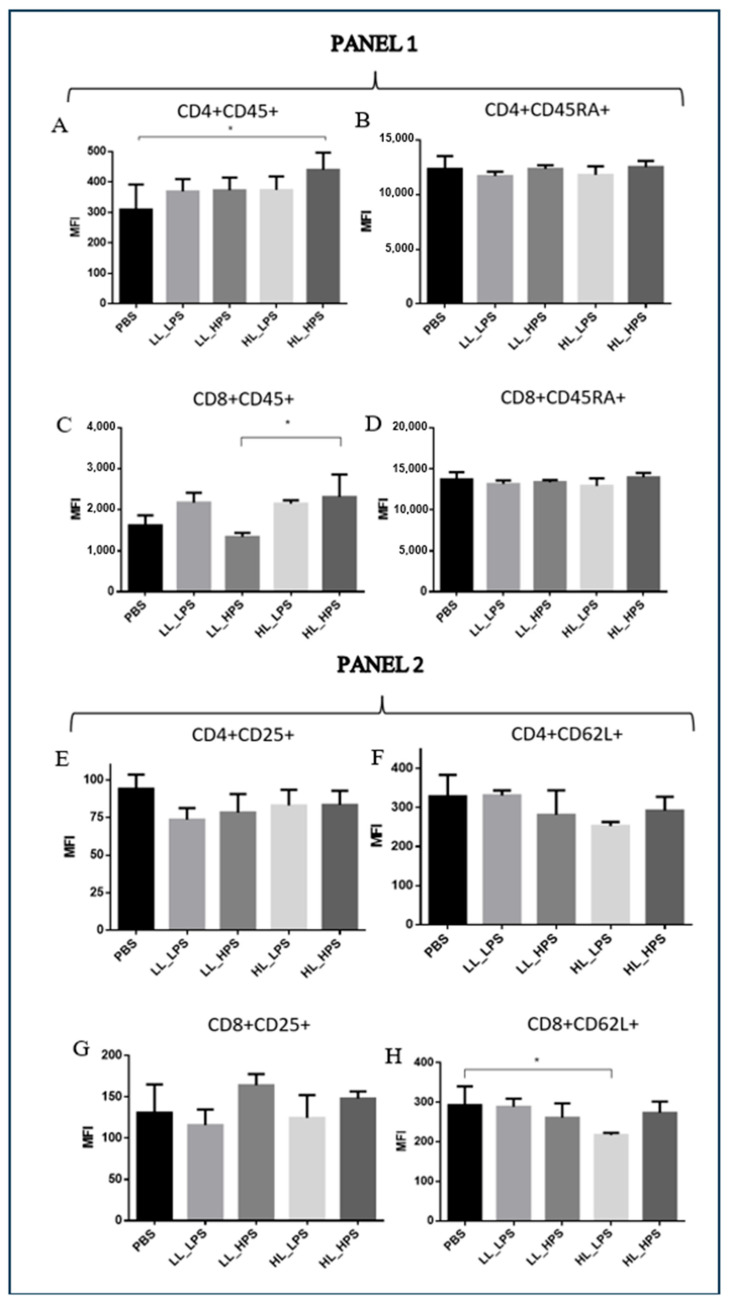
Flow cytometry analyses of the mice splenic CD4+ and CD8+ T populations. PANEL 1: markers for CD45+ and CD45RA+ T cells. MFI of (**A**) CD4+CD45+; (**B**) CD4+CD45RA+; (**C**) CD8+CD45+; (**D**) CD8+CD45RA+. PANEL 2: (**E**) CD4+CD25+; (**F**) CD4+CD62L+; (**G**) CD8+CD25+; (**H**) CD8+CD62L+; MFI: mean fluorescence intensity. (n = 3). One-way ANOVA followed by Tukey’s post hoc test was performed for statistical analysis; * *p* < 0.05. Data are presented as the mean ± SD.

**Figure 8 pharmaceutics-16-01177-f008:**
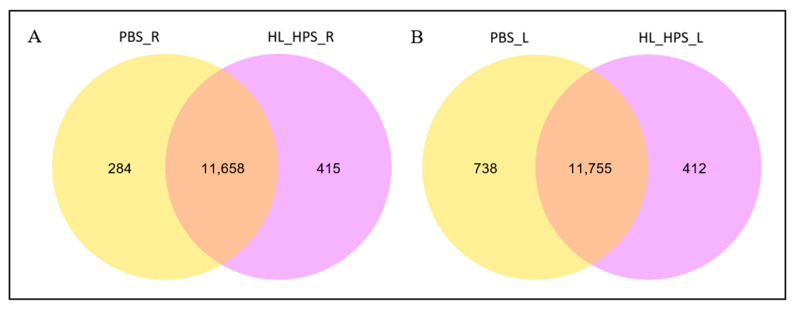
Venn diagram of the numbers and distribution of unique and co-expressed genes for the PBS and HL_HPS treatment conditions. (**A**) Right tumors. (**B**) Left tumors.

**Figure 9 pharmaceutics-16-01177-f009:**
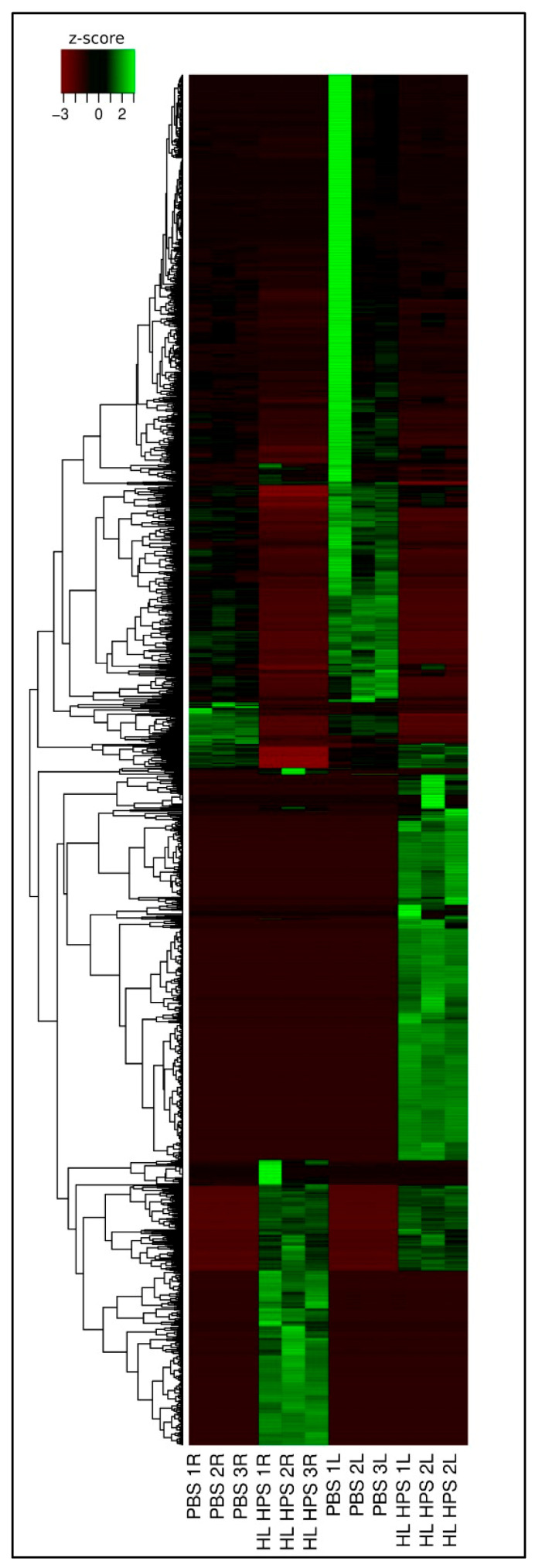
Heatmap of differentially expressed genes between the right and left tumor samples in PBS and HL_HPS groups. The heatmap displays the Z-score normalized differential expression across three replicates. R indicates right tumor, and L indicates left tumor.

**Figure 10 pharmaceutics-16-01177-f010:**
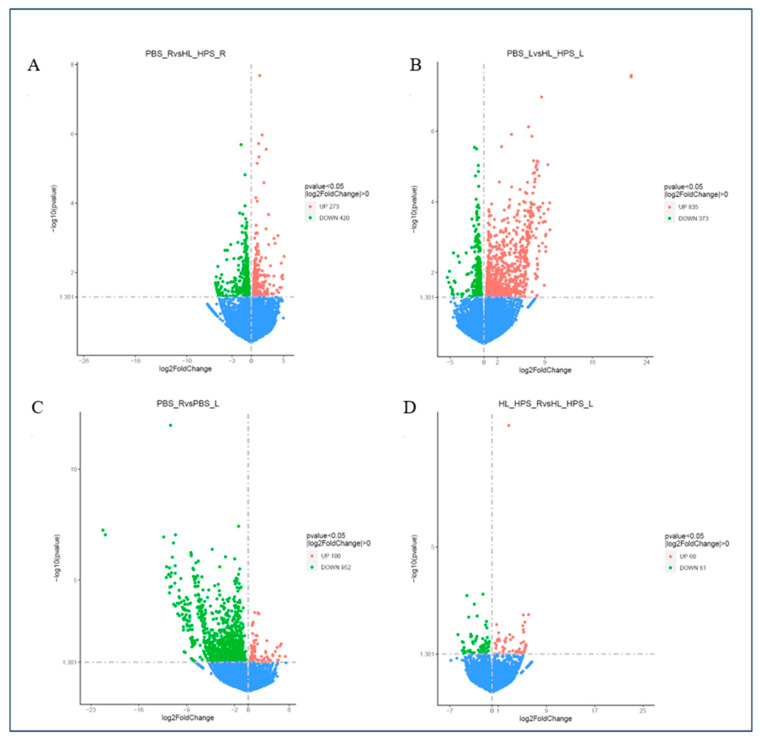
Volcano plot of differential expression of genes from RNA-Seq comparing the (**A**) PBS_R vs. HL_HPS_R; (**B**) PBS_L vs. HL_HPS_L; (**C**) PBS_R vs. PBS_L; (**D**) HL_HPS_R vs. HL_HPS_L. Each dot indicates one gene. Red dots represent upregulated genes, while green dots represent downregulated genes.

**Figure 11 pharmaceutics-16-01177-f011:**
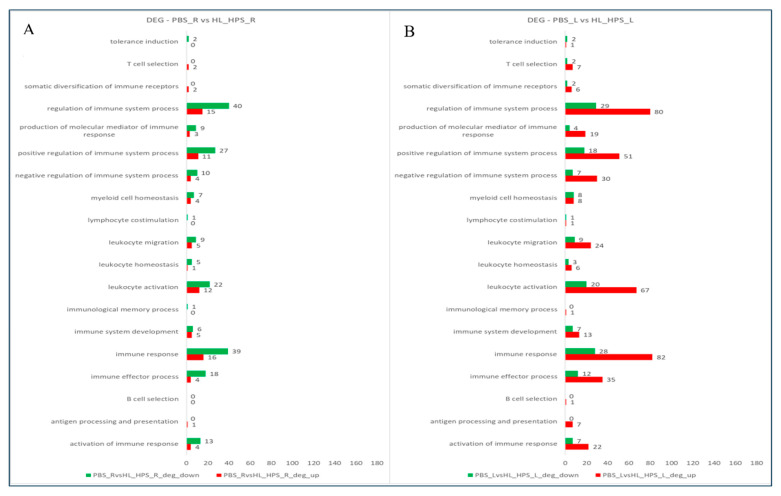
Number of upregulated (red) and downregulated (green) genes based on Gene Ontology (GO) in Immune System Processes (GO:0002376). (**A**) Comparison between PBS_R and HL_HPS_R. (**B**) PBS_L vs. HL_HPS_L.

**Figure 12 pharmaceutics-16-01177-f012:**
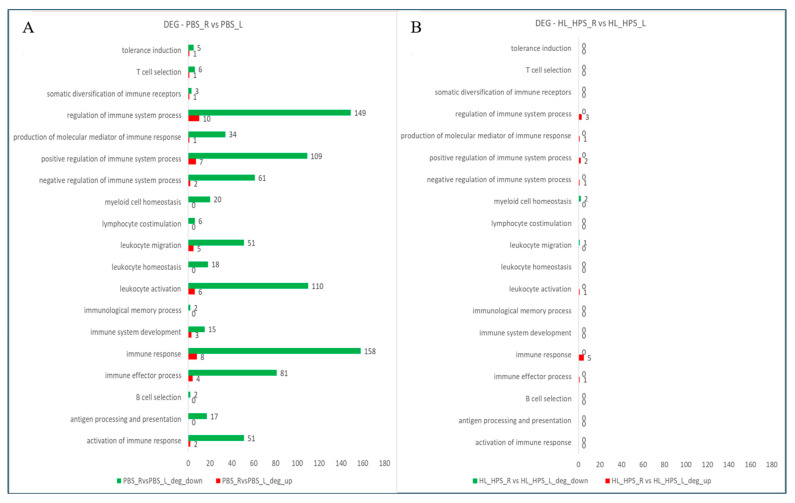
Number of upregulated (red) and downregulated (green) DEGs based on Gene Ontology (GO) in Immune System Processes (GO:0002376). (**A**) Comparison between PBS_R and PBS_L. (**B**) HL_HPS_R vs. HL_HPS_L.

**Table 1 pharmaceutics-16-01177-t001:** Results of erythrogram, leukogram, and platelet count of female C57bl6 on day 18 after treatment with PDT or PBS. RBC—red blood cells; HGB—hemoglobin; HCT—hematocrit; MCV—mean corpuscular volume; MCH—mean corpuscular hemoglobin; MCHC—mean corpuscular hemoglobin concentration; RDW-CV—red cell distribution width coefficient of variation; RDW-SD—red cell distribution width standard deviation; WBC—total white blood cell count; NEU—neutrophils; LYM—lymphocytes; MON—monocytes; EOS—eosinophils; BAS—basophils; PLT—platelets; MPV—mean platelet volume; PDW—platelet distribution width; PCT—plateletcrit; g/dL—grams per deciliter; fL—fentoliters; pg—picograms. Data correspond to mean ± standard deviation (SD).

	PBS	LL_LPS	LL_HPS	HL_LPS	HL_HPS
Erythrogram					
RBC (×10^3^/µL)	6.6 ± 1.1	4.0 ± 1.7	5.6 ± 1.5	4.4 ± 2.6	3.4 ± 1.3
HGB (g/dL)	10.9 ± 1.6	7.0 ± 2.7	9.1 ± 2.3	7.1 ± 4.1	5.8 ± 2.2
HCT (%)	31.3 ± 4.7	19.8 ± 8.1 a	27.0 ± 7.6	20.8 ± 11.9	17.6 ± 7.1 a
MCV (fL)	47.3 ± 2.0	50.2 ± 4.3	48.0 ± 1.0	48.5 ± 8.1	52.1 ± 2.8
MCH (pg)	16.5 ± 0.4	17.7 ± 1.0	16.4 ± 0.2	16.3 ± 0.4	17.2 ± 0.2
MCHC (g/dL)	35.0 ± 0.1	35.4 ± 1.4	34.0 ± 1.0	34.4 ± 4.7	33.1 ± 2.0
RDW-CV (%)	16.1 ± 3.8	19.6 ± 6.7	19.9 ± 5.4	24.6 ± 7.7	23.0 ± 5.6
RDW-SD (fL)	34.4 ± 10.3	44.4 ± 18.8	42.9 ± 13.3	54.7 ± 22.8 a	54.3 ± 15.6 a
Leukogram					
WBC (×10^3^/µL)	10.8 ± 2.7	4.9 ± 1.3 a	5.1 ± 1.5 a	5.0 ± 0.2 a	4.4 ± 1.7 a
NEU (×10^3^/µL)	3.2 ± 0.9	1.9 ± 0.7 a	1.4 ± 0.7 a	1.2 ± 0.4 a	1.4 ± 0.6 a
LYM (×10^3^/µL)	6.2 ± 1.3	2.5 ± 1.0 a	3.1 ± 0.7 a	3.0 ± 0.8 a	2.7 ± 1.4 a
MON (×10^3^/µL)	1.4 ± 0.7	0.5 ± 0.1 a	0.4 ± 0.5	0.6 ± 0.4	0.3 ± 0.2 a
EOS (×10^3^/µL)	0.1 ± 0.0	0.0 ± 0.0	0.2 ± 0.2	0.1 ± 0.1	0.1 ± 0.0
BAS (×10^3^/µL)	0.0 ± 0.0	0.0 ± 0.0	0.0 ± 0.0	0.0 ± 0.0	0.0 ± 0.0
NEU (%)	29.7 ± 3.7	39.5 ± 11.3	25.2 ± 7.7	24.2 ± 7.8	31.1 ± 9.4
LYM (%)	59.2 ± 7.0	49.6 ± 9.8	65.4 ± 14.3	60.6 ± 15.9	60.4 ± 13.1
MON (%)	10.2 ± 5.3	10.0 ± 2.4	7.1 ± 6.1	12.3 ± 7.8	6.7 ± 3.4
EOS (%)	0.9 ± 0.3	0.9 ± 0.8	2.4 ± 2.6	2.9 ± 1.6	1.9 ± 0.7
BAS (%)	0.0 ± 0.0	0.0 ± 0.0	0.0 ± 0.0	0.0 ± 0.0	0.0 ± 0.0
Platelet parameters					
PLT (×10^3^/µL)	682.6 ± 107.0 a	509.8 ± 187.9	476.8 ± 293.4	624.0 ± 308.0	530.8 ± 59.2 a
MPV (fL)	5.5 ± 0.5	6.2 ± 1.0	5.4 ± 0.2	5.8 ± 0.2	6.6 ± 0.5
PDW	15.8 ± 0.4	15.9 ± 0.4	15.8 ± 0.2	16.0 ± 0.6	16.3 ± 0.4
PCT (%)	0.4 ± 0.1	0.3 ± 0.1	0.3 ± 0.2	0.4 ± 0.2	0.4 ± 0.1

a = Significant difference compared to PBS group, detected by Tukey’s multiple-comparison test, *p* < 0.01.

**Table 2 pharmaceutics-16-01177-t002:** List of DEGs upregulated and downregulated in HL_HPS_R in PBS_R vs. HL_HPS_R and HL_HPS_L in PBS_L vs. HL_HPS_L based on Gene Ontology (GO) in Immune System Processes (GO:0002376). (|log2(FoldChange)| ≥ 1 and padj ≤ 0.05).

	DEGs_UP	DEGs_DOWN
PBS_R vs. HL_HPS_R	Asxl1, Brpf1, Cacnb4, Cd3d, Chid1, Cmtm7, Cnpy3, Crtam, Cxcl11, Gli3, Gpr17, H2-Eb2, Herc6, Kat7, Kit, Klrk1, Mef2c, Nuggc, Prss56, Rbm15, Shld3, Slc25a38, Socs1, Sox6, Spp1, Stxbp4, Tab1, Traf4.	Ackr3, Acvr1b, Ahr, Arg1, C6, Ccnd3, Cd36, Cdc42ep2, Cdkn1a, Cebpg, Ctla2a, Defb1, Dusp10, Ecm1, Egr3, Emp2, Ephb3, Fosl2, Fzd9, Gprc5b, Gramd4, Grem1, Hcfc2, Hexim1, Il36g, Ivl, Jun, Jund, Kcnj8, Lrp1, Mavs, Meis1, Mitf, Mospd2, Nfkb2, Nos2, Nr4a3, Pde4b, Plec, Prkd2, Ptk6, Pura, Pvr, Rarg, Sfn, Sh2b2, Six1, Sox9, Tbkbp1, Thra, Tmem45b, Tnfaip3, Tnfsf14, Tnfsf9, Trpm4, Ttbk1, Vegfa, Xkr8, Zbtb7a, Zbtb7b.
PBS_L vs. HL_HPS_L	Adgre1, Adgrf4, Adtrp, Alcam, Angpt1, Aqp3, Bank1, Bcl11a, Bcl11b, Bcl2a1d, Blnk, Bmi1, Bmp4, Bpifc, Ccl28, Ccn3, Cd180, Cd209b, Cd55, Ciita, Coch, Crhr1, Ctsc, Ctse, Cx3cl1, Cx3cr1, Cxadr, Cxcl10, Cxcr6, Cyld, Dapk2, Dapl1, Ddx3x, Defb1, Dhx15, Dnase1l3, Dock11, Edn2, Egr3, Emp2, Endou, Erap1, Evpl, Ext1, F2rl1, F830016B08Rik, Fgfr3, Fgl2, Foxn1, Foxp1, Fzd5, Gata3, Gbp10, Gbp6, Herc6, Igf1, Ighm, Il1rap, Irf4, Itgal, Itgax, Itgb8, Ivl, Kat7, Kif5b, Lacc1, Lair1, Lcp2, Lef1, Ly6d, Marchf7, Mecom, Mef2c, Mill1, Mysm1, Nedd9, Nfkbiz, Notch1, Padi4, Pag1, Pdgfd, Pik3ap1, Pla2g2f, Pparg, Ppl, Prdm1, Prkcb, Prlr, Psg17, Ptger4, Ptprc, Pycard, Rasgrp1, Rbm15, Rel, Ret, Rnf115, S100a14, Selp, Serpinb9, Sfn, Sfrp1, Shld3, Sirpb1a, Sirt1, Skil, Slamf8, Slc40a1, Socs6, Sp1, Sp3, Spn, Tbx1, Tcea1, Trim29, Usp14, Usp9x, Vav3, Wdfy4, Wnt5a, Ythdf2, Zbtb6.	Aire, Ccr10, Cdc42ep2, Cebpg, Cxcl1, Cybc1, Cyren, Ephb4, F7, Fkbp1b, Fzd9, Gata2, Gdf15, Gpr137, Gprc5b, Hexim1, Hs1bp3, Il15, Il17d, Irf3, Jmjd6, Kcnj8, Kcnn4, Kmt5c, Mark4, Men1, Mitf, Mospd2, Nemp1, Nfkb2, Nr4a3, Orai1, Ppbp, Prdm16, Prelid1, Prkd2, Rac3, Rara, Rarg, Rbm14, Slc37a4, Sppl2b, Tal1, Tbkbp1, Trim68, Tusc2, Vegfa, Xkr8, Zbtb7b.

**Table 3 pharmaceutics-16-01177-t003:** List of DEGs upregulated and downregulated in PBS_L in PBS_R vs. PBS_L and HL_HPS_L in HL_HPS_R vs. HL_HPS_L based on Gene Ontology (GO) in Immune System Processes (GO:0002376). (|log2(FoldChange)| ≥ 1 and padj ≤ 0.05).

	DEGs_UP	DEGs_DOWN
PBS_R vs. PBS_L	Aire, Cdh17, Evl, Gpr137, Kcnj8, Kcnn4, Kmt5c, Ppbp, Prss56, Prxl2a, Rara, Vegfb	Acp5, Adgre1, Adgrf4, Adipoq, Adtrp, Aim2, Alox5, Ang, Angpt1, Anxa3, Aqp3, Bcl11b, Bcl2a1d, Blnk, Bmp4, Bpifc, Bst1, Btk, C1qb, C1qc, C3ar1, C6, Camk1d, Ccl21a, Ccl6, Ccl7, Ccl8, Ccl9, Ccn3, Ccr2, Ccr5, Cd209b, Cd209d, Cd244a, Cd24a, Cd36, Cd37, Cd55, Cd84, Cd86, Cebpa, Cfh, Clec2d, Clec2g, Clec4a1, Clec4a2, Clec4a3, Cmklr1, Coch, Coro1a, Crhr1, Csf1r, Csf2rb, Csf2rb2, Ctsc, Ctse, Ctss, Cxcl12, Cxcl14, Cxcl16, Cybb, Cysltr1, Dapk1, Dapk2, Dapl1, Defb1, Dock2, Dpp4, Dtx4, Ear2, Edn2, Ednrb, Egr1, Egr3, Emp2, Ephb3, Evpl, F2rl1, Fcer1g, Fcgr2b, Fcgr3, Fcgr4, Fgfr3, Fgl2, Fgr, Foxn1, Gas6, Gata3, Gbp2b, Gm5431, Gpr55, H2-T24, Hck, Hspb1, Ifi204, Ifi205, Ifi207, Ifi209, Igf1, Ighm, Ikzf1, Il1rl2, Il4ra, Il6ra, Inpp5d, Irf5, Itgal, Itgam, Itgb2, Ivl, Kitl, Lair1, Laptm5, Lcp2, Lef1, Lfng, Lilrb4a, Lilrb4b, Lmo2, Lpxn, Ly6d, Ly86, Lyn, Lyve1, Mafb, Marchf1, Mill1, Mpeg1, Mrgprb1, Myo1f, Myo1g, Mysm1, Naip6, Ncf1, Nckap1l, Nedd9, Nfam1, Notch1, P2ry14, Padi4, Pag1, Pck1, Pdgfd, Pik3ap1, Pirb, Pla2g2f, Plcl2, Pld4, Plscr2, Pou2f2, Pparg, Ppl, Prdm1, Prkcb, Prkch, Prlr, Psg17, Ptafr, Ptger4, Ptprc, Ptpre, Ptpro, Rac2, Rasgrp1, Rftn1, Rnase4, S100a14, S1pr1, Sash3, Selp, Sema4a, Sfn, Sfrp1, Sirpb1c, Slamf7, Slc11a1, Slc7a2, Slfn1, Slfn2, Spink5, Svep1, Tbx1, Themis2, Tifab, Tlr13, Tlr8, Tlr9, Tmem229b, Tmem98, Tnfaip8l2, Tnfrsf13b, Tnfrsf1b, Tnfrsf21, Tnip3, Trem2, Trim29, Trpm2, Tyrobp, Vav1, Vcam1, Vsig4, Vsir, Wdfy4, Wfdc17, Wnt10b, Zeb1
HL_HPS_R vs. HL_HPS_L	Egr3, Endou, Il36g, Il36rn, Klk5, Ptk6, Sox9	Dpep1, Hba-a1, Hba-a2

## Data Availability

The data presented in this study are available in this article.

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
