# Peer review of "Direct and Abscopal Antitumor Responses Elicited by AlPcNE-Mediated Photodynamic Therapy in a Murine Melanoma Model"

_pharmaceutics, 2024, doi:10.3390/pharmaceutics16091177_

Round 1

Reviewer 1 Report

Comments and Suggestions for Authors

The article titled "Direct and Abscopal Antitumor Responses Elicited by AlPcNE-Mediated Photodynamic Therapy in a Murine Melanoma Model" presented by Morais et al. in Pharmaceutics aims to evaluate different PDT protocols with AlPcNE in a murine melanoma model, both in the primary irradiated tumor and in a distant tumor, to determine a systemic immunomodulatory effect. The article presents logically organized in vivo results. However, I believe that some aspects need to be better discussed, and additional experiments will be required to reach the resulting conclusions.

  1. The introduction section could be enriched by presenting other studies that have addressed the topic of melanoma treatment with PDT and the nanosystem proposed in this article: https://doi.org/10.3390/pharmaceutics14112474; https://doi.org/10.3390/pharmaceutics16070941.
  2. Specifically, this statement "However, the antitumor potential in the treatment of melanoma and a possible effect in the activation of the immune system remains to be investigated" requires a deeper exploration of the state of the art of studies that have addressed the topic. For example, the authors have published a recent work on the subject: https://doi.org/10.3390/pharmaceutics16070941.
  3. Please describe in greater detail the PDT protocol used with the animals. It is not clear how the illumination of the tumor area was performed. Was an optical fiber used? Was the protocol conducted under anesthesia?
  4. Regarding the results section, it is unclear what figure 2B shows, what AUC is, and how it was calculated. Please provide a reference regarding its use and interpretation.
  5. The following statement on page 5 requires experimental support: "This can be due to the lower oxidation of cell components in the tumors treated with the LL_LPS protocol, as it involved lower concentrations of both AlPcNE and light energy." Was the amount of photosensitizer in the tumor quantified, and was its relationship to tumor mass considered to allow comparisons between tumors of different sizes?
  6. The caption for figure 3 describes other images that have not been included in the image panel. On the other hand, it would be interesting to perform an immunohistochemical analysis to determine immune infiltration and its differences between the different experimental groups. This would increase the possibility of a more robust and reliable analysis of the results presented with RNA-seq.
  7. An analysis of the immune infiltration of both primary and distant tumors similar to that performed by flow cytometry on the spleens is required to determine and complement the RNA-seq studies.
  8. Please provide a deeper discussion comparing with other studies that have evaluated the potential of different PDT therapeutic regimens on tumors and their microenvironment. These articles could help: https://doi.org/10.3389/fonc.2019.00811; https://doi.org/10.3390/cells12111541; https://doi.org/10.1016/j.jphotobiol.2019.111596.

Author Response

  1. The introduction section could be enriched by presenting other studies that have addressed the topic of melanoma treatment with PDT and the nanosystem proposed in this article: https://doi.org/10.3390/pharmaceutics14112474; https://doi.org/10.3390/pharmaceutics16070941.

Answer: Thank you for suggesting the inclusion of additional studies in the introduction section. As recommended, we have reviewed and incorporated references from the suggested articles, as seen in lines 70-78.

  1. Specifically, this statement "However, the antitumor potential in the treatment of melanoma and a possible effect in the activation of the immune system remains to be investigated" requires a deeper exploration of the state of the art of studies that have addressed the topic. For example, the authors have published a recent work on the subject: https://doi.org/10.3390/pharmaceutics16070941.

Answer: Thank you for your insightful feedback. We acknowledge the need for a change in this sentence and have made the suggested modification in lines 78–80.

  1. Please describe in greater detail the PDT protocol used with the animals. It is not clear how the illumination of the tumor area was performed. Was an optical fiber used? Was the protocol conducted under anesthesia?

Answer: Thank you for your valuable feedback. We have revised the manuscript to include a more detailed description of the photodynamic therapy (PDT) protocol in section 2.4.

  1. Regarding the results section, it is unclear what figure 2B shows, what AUC is, and how it was calculated. Please provide a reference regarding its use and interpretation.

Answer: Thank you for your feedback. Figure 2B illustrates the relationship between treatment groups and their corresponding Area Under the Curve (AUC) values, representing the overall response over time. The AUC was calculated by integrating the response curve to provide a comprehensive measure of the treatment effect. We will clarify this in the legend to Figure 2B. As a suggested reference for this analysis, we recommend the work by Matthews JN, Altman DG, Campbell MJ, Royston P. Analysis of serial measurements in medical research. BMJ. 1990 Jan 27;300(6719):230-5. doi: 10.1136/bmj.300.6719.230. PMID: 2106931; PMCID: PMC1662068. 

  1. The following statement on page 5 requires experimental support: "This can be due to the lower oxidation of cell components in the tumors treated with the LL_LPS protocol, as it involved lower concentrations of both AlPcNE and light energy." Was the amount of photosensitizer in the tumor quantified, and was its relationship to tumor mass considered to allow comparisons between tumors of different sizes?

Answer: Thank you for your question. The statement regarding the lower oxidation of cell components in tumors treated with the LL_LPS protocol is based on the fact that this protocol involved a lower concentration of the photosensitizer (AlPcNE) applied directly to the tumor, as well as a lower dose of light energy. Although we did not quantify the amount of photosensitizer within the tumors or account for tumor mass in our analysis, the rationale for the observed effect is supported by the lower input of both key elements (photosensitizer and light) that drive the photodynamic response.

  1. The caption for figure 3 describes other images that have not been included in the image panel. On the other hand, it would be interesting to perform an immunohistochemical analysis to determine immune infiltration and its differences between the different experimental groups. This would increase the possibility of a more robust and reliable analysis of the results presented with RNA-seq.

Answer: Thank you for your careful review and the observation regarding Figure 3. We apologize for the discrepancy in the figure caption, which mentioned images not included in the panel. We have revised the caption to accurately describe the images presented. Regarding your suggestion to perform immunohistochemical analysis to determine immune infiltration, we agree that this would provide valuable insights into the immune response within the tumor microenvironment. At present, we have not conducted immunohistochemical staining for immune cell markers, but we recognize the importance of this analysis in complementing our RNA-seq results. We are currently planning to perform this analysis in future studies to enhance the robustness and reliability of our findings.

  1. An analysis of the immune infiltration of both primary and distant tumors similar to that performed by flow cytometry on the spleens is required to determine and complement the RNA-seq studies.

Answer: We appreciate your suggestion to analyze immune infiltration in both primary and distant tumors using a method similar to the flow cytometry performed on spleens. This analysis would indeed provide a more comprehensive understanding of the immune landscape across different tissues and better complement our RNA-seq data. Unfortunately, due to the limitations of our current study design and available resources, we were unable to perform flow cytometry on tumor samples. 

  1. Please provide a deeper discussion comparing with other studies that have evaluated the potential of different PDT therapeutic regimens on tumors and their microenvironment. These articles could help: https://doi.org/10.3389/fonc.2019.00811; https://doi.org/10.3390/cells12111541; https://doi.org/10.1016/j.jphotobiol.2019.111596.

Answer: Thank you for highlighting the need for a more comprehensive discussion of our findings within the context of the existing literature on PDT regimes and their effects on tumors and the tumor microenvironment. We have reviewed the suggested articles https://doi.org/10.3389/fonc.2019.00811 and https://doi.org/10.1016/j.jphotobiol.2019.111596 and incorporated relevant comparisons and discussions into our manuscript, as reflected in lines 421-428.

Reviewer 2 Report

Comments and Suggestions for Authors

The manuscript submitted for review describes very interesting results regarding phototherapy of cancer. However, I have a few comments about this manuscript:

1. the Authors should describe the tumor photodestruction experiments in more detail (in the Materials and methods subsection), with particular emphasis on the photosensitizer concentrations used (these data are not included in the materials and methods).

2. the Authors should correct the text in terms of the spelling of subscripts and superscripts (e.g. chemical formula of carbon dioxide).

3. Can the doses of light used be painful for the patient?

Comments on the Quality of English Language

The English language requires only minor corrections.

Author Response

  1. the Authors should describe the tumor photodestruction experiments in more detail (in the Materials and methods subsection), with particular emphasis on the photosensitizer concentrations used (these data are not included in the materials and methods).

Answer: Thank you for pointing out the need for a more detailed description of the tumor photodestruction experiments. We have revised the Materials and Methods section to include comprehensive information regarding the concentration of the photosensitizer used in the experiments.

  1. the Authors should correct the text in terms of the spelling of subscripts and superscripts (e.g. chemical formula of carbon dioxide).

Answer: We appreciate your attention to detail regarding the formatting of subscripts and superscripts. We have thoroughly reviewed the manuscript and corrected any inconsistencies, particularly with chemical formulas.

  1. Can the doses of light used be painful for the patient?

Answer: Thank you for raising the important issue of patient comfort during photodynamic therapy (PDT). The light doses used in our study were chosen based on their efficacy in inducing tumor photodestruction in preclinical models, where the animals were under general anesthesia during the application to ensure there was no discomfort. However, we recognize that translating these doses to clinical settings requires careful consideration of patient safety and comfort. It is known that high-intensity light exposure can cause discomfort or even pain due to thermal effects, depending on the photosensitizer and treatment parameters used. In clinical practice, pain management strategies are often employed, including the use of local anesthesia, cooling devices, or adjusting light dosages.

Reviewer 3 Report

Comments and Suggestions for Authors

The authors present a very interesting study about the application of phthalocyanine based nanoemulsion for the treatment of melanoma. The authors present very interesting findings also regarding the systemic effect of the treatment NE formulation allied with PDT, showing that high level of light dose and the higher concentration of photosensitizer NE can reduce even a secondary tumor growth not treated directly. The study is very complete with results of blood analysis, showing the systemic effect of the treatments as well as flow cytometry studies for the analysis of T lymphocyte populations. Lastly, authors bring the RNA-Seq analysis showing that tumor growth inhibition mechanism was different for the primary and secondary tumor sites. I address below some suggestions to be considered by the authors.

Line 35

In the abstract there is an abbreviation not explained: “AlPcNE”. Please, if possible, describe briefly the meaning of this abbreviation.

Lines 64-65

What the authors mean with the phrase: “AlPcNE, in particular, presents high 64 activity in aqueous media”. Please write more clearly about this activity in the text.

Line 94

Figure 1 is a very nice image describing the experimental planning. Please put the indication that it is in days, as it is missing this information in the figure.

Line 97

The nanoemulsion containing the photosensitizer employed in this research was previously published, as indicated by the authors, with details regarding preparation and physicochemical characterization. It would be interesting to add in the introduction a paragraph or a phrase summarizing the main findings and characterization of the selected formulation (size, concentration of aluminium-phthalocyanine chloride).

Lines 164-165

To correct the light dose shows unit “J/cm2”.

Author Response

  1. Line 35 In the abstract there is an abbreviation not explained: “AlPcNE”. Please, if possible, describe briefly the meaning of this abbreviation.

 Answer: Thank you for pointing out the unexplained abbreviation. We have revised the abstract to include a brief explanation of “AlPcNE”

  1. Lines 64-65 What the authors mean with the phrase: “AlPcNE, in particular, presents high 64 activity in aqueous media”. Please write more clearly about this activity in the text.

 Answer: We appreciate your request for clarification on this phrase. To make the text clearer, we have revised it to provide more detail about the specific activity we are referring to:

The phrase has been revised to: “AlPcNE, in particular, exhibits significantly higher photodynamic activity in aqueous media compared to its non-nanoemulsified form, AlPc.”

  1. Line 94 Figure 1 is a very nice image describing the experimental planning. Please put the indication that it is in days, as it is missing this information in the figure.

 Answer: Thank you for your positive feedback on Figure 1 and for pointing out the missing information. We have updated Figure 1 to clearly indicate that the timeline is in days. 

  1. Line 97 The nanoemulsion containing the photosensitizer employed in this research was previously published, as indicated by the authors, with details regarding preparation and physicochemical characterization. It would be interesting to add in the introduction a paragraph or a phrase summarizing the main findings and characterization of the selected formulation (size, concentration of aluminium-phthalocyanine chloride).

 Answer: Thank you for your valuable suggestion. To provide a clearer understanding of the nanoemulsion formulation used in this study, we have added a summary of its key physicochemical characteristics to the introduction. The nanoemulsion containing aluminum-phthalocyanine chloride (AlPc) used in this research was carefully characterized, revealing a mean hydrodynamic diameter (DH) of 27.15 nm ± 1.070, a polydispersity index (PdI) of 0.178 ± 0.026, and a zeta potential (ZP) of -0.340 ± 0.188 mV. The final concentration of AlPcNEin the nanoformulation (AlPcNE) was 40 µM.

  1. Lines 164-165 To correct the light dose shows unit “J/cm2”.

Answer: Thank you for identifying the need to correct the light dose units. We have reviewed the manuscript and corrected the units for light dose to “J/cm²” to ensure consistency and accuracy in scientific notation throughout the text.

Reviewer 4 Report

Comments and Suggestions for Authors

This is an interesting report that describes direct and indirect effects of PDT on melanoma. A few modifications are suggested. Line 54 refers to ‘a specific light wavelength’. All wavelengths are ‘specific’. The wavelength must correspond to an absorbance band of the photosensitizer. Since melanomas tend to be highly pigmented. The wavelength should be in the far red or near IR for there to be adequate tissue penetration. ‘Cytotoxicity’ (line 54) is an ambiguous term. What is needed is tumor eradication, either via direct or indirect effects. Line 62 should indicate that the wavelength is also important because of light scattering effects which limit efficacy. 

The mode of drug treatment is not clear. How was the photosensitizer administered? Tail-vein injections represent the closest approximation to what is done in clinical studies. While the plan is to evaluate effects on tumor that is not irradiated, it appears that a tail-vein injection could be used. The authors need to clarify this point.

It appears from Fig. 2 that only the low light:low drug protocol was ineffective. It was the high light:high drug protocol that produced an optimal effect on the remote tumor. Other variations had some effect. Since PDT is usually believed to be effective only in combination with light, this report is interesting since it indicates that immunologic effects are also relevant. It can be difficult to irradiate all sites of neoplasia by direct photokilling because these sites are either unknown or not reached by a sufficient level of irradiation (because of light-scattering or other effects). 

Lines 456-461 discuss the inability of immunologic effects to lead to tumor eradication. Fig. 2 appears to indicate that while tumor growth is suppressed by some protocols, the tumor is never eradicated. Tumors continue to grow but at a reduced rate. Even the optimally treated animals to not appear to be ‘cured’. And while there are anti-tumor effects on the remote tumor (Fig. 4), tumor growth does continue.  This report does, however, show that there are significant effects of PDT on tumor that was not irradiated. 

 Minor point: It was well-known that PDT involved effects involving reactive oxygen species long before the work of Pan et al (2021).    

Author Response

This is an interesting report that describes direct and indirect effects of PDT on melanoma. A few modifications are suggested. Line 54 refers to ‘a specific light wavelength’. All wavelengths are ‘specific’. The wavelength must correspond to an absorbance band of the photosensitizer. Since melanomas tend to be highly pigmented. The wavelength should be in the far red or near IR for there to be adequate tissue penetration. ‘Cytotoxicity’ (line 54) is an ambiguous term. What is needed is tumor eradication, either via direct or indirect effects. Line 62 should indicate that the wavelength is also important because of light scattering effects which limit efficacy. 

 The mode of drug treatment is not clear. How was the photosensitizer administered? Tail-vein injections represent the closest approximation to what is done in clinical studies. While the plan is to evaluate effects on tumor that is not irradiated, it appears that a tail-vein injection could be used. The authors need to clarify this point.

 It appears from Fig. 2 that only the low light:low drug protocol was ineffective. It was the high light:high drug protocol that produced an optimal effect on the remote tumor. Other variations had some effect. Since PDT is usually believed to be effective only in combination with light, this report is interesting since it indicates that immunologic effects are also relevant. It can be difficult to irradiate all sites of neoplasia by direct photokilling because these sites are either unknown or not reached by a sufficient level of irradiation (because of light-scattering or other effects). 

 Lines 456-461 discuss the inability of immunologic effects to lead to tumor eradication. Fig. 2 appears to indicate that while tumor growth is suppressed by some protocols, the tumor is never eradicated. Tumors continue to grow but at a reduced rate. Even the optimally treated animals to not appear to be ‘cured’. And while there are anti-tumor effects on the remote tumor (Fig. 4), tumor growth does continue.  This report does, however, show that there are significant effects of PDT on tumor that was not irradiated. 

 Minor point: It was well-known that PDT involved effects involving reactive oxygen species long before the work of Pan et al (2021).    

Thank you for your thoughtful feedback and suggestions regarding our manuscript. We appreciate the opportunity to address your comments and have made the necessary revisions to improve the clarity and quality of the report. Below are our responses to your specific points:

Answer: About wavelength: We agree that referring to “a specific light wavelength” can be clarified. We have revised the text to specify that the wavelength should match the absorbance band of the photosensitizer, according to lines 53-57.

About Line 54:

Thank you for pointing out the ambiguity in the use of the term “cytotoxicity.” We agree that the ultimate goal is tumor eradication, which can be achieved through both direct and indirect effects. In our context, "cytotoxicity" refers to the ability of the treatment to induce cell death in tumor cells, which is a critical step towards tumor eradication. However, to avoid any confusion, we have revised the text to more accurately reflect the intended outcome, emphasizing the goal of complete tumor elimination through the mechanisms employed in our study. 

About Photosensitizer Administration:

  • We have clarified the mode of photosensitizer administration in the Methods section. 

About ROS reference:

  • Thank you for pointing out this minor issue. We have updated the reference to Kolarova et al. (2007) to better reflect the well-established understanding of this mechanism.

Round 2

Reviewer 1 Report

Comments and Suggestions for Authors

I do not have more comments. Corrections and explanation seem to be proper.